

# A Cloud-Ozone Data Product from Aura OMI and MLS Satellite Measurements

Jerald R. Ziemke[1,2], Sarah A. Strode[2,3], Anne R. Douglass[2], Joanna Joiner[2], Alexander Vasilkov[2,4], Luke D. Oman[2], Junhua Liu[2,3], Susan E. Strahan[2,3], Pawan K. Bhartia[2], David P. Haffner[2,4]

[1]Morgan State University, Baltimore, Maryland, USA

[2]NASA Goddard Space Flight Center, Greenbelt, Maryland, USA

[3]Universities Space Research Association, Columbia, MD, USA

[4]SSAI, Lanham, Maryland, USA

**Abstract.** Ozone within deep convective clouds is controlled by several factors involving photochemical reactions and transport. Gas-phase photochemical reactions, and heterogeneous surface chemical reactions involving ice, water particles, and aerosols inside the clouds all contribute to the distribution and net production and loss of ozone. Ozone in clouds is also dependent on convective transport that carries low troposphere/boundary layer ozone and ozone precursors upward into the clouds. Characterizing ozone in thick clouds is an important step for quantifying relationships of ozone with tropospheric $H_2O$, OH production, and cloud microphysics/transport properties. Although measuring ozone in deep convective clouds from either aircraft or balloon ozonesondes is largely impossible due to extreme meteorological conditions associated with these clouds, it is possible to estimate ozone in thick clouds using backscattered solar UV radiation measured by satellite instruments. Our study combines Aura Ozone Monitoring Instrument (OMI) and Microwave Limb Sounder (MLS) satellite measurements to generate a new research product, monthly-mean ozone concentrations in deep convective clouds between 30°S to 30°N for October 2004 – April 2016. These measurements reveal key features of cloud ozone including: persistent low ozone concentrations in the tropical



Pacific of ~10 ppbv or less; concentrations of up to 60 pphv or greater over landmass regions of
South America, southern Africa, Australia, and India/east Asia; connections with tropical ENSO
events; and intra-seasonal/Madden-Julian Oscillation variability.  Analysis of OMI aerosol
measurements suggests a cause and effect relation between boundary layer pollution and
elevated ozone inside thick clouds over land-mass regions including southern Africa and
India/east Asia.

**1. Introduction.**

Measuring tropospheric ozone in deep convective clouds including convective outflow regions in
the mid-upper troposphere is important for several reasons.  Ozone in the upper troposphere is a
major greenhouse gas that contributes to climate forcing.  The IPCC 2013 Report (e.g., in
Hartmann et al., 2014; http://www.ipcc.ch/report/ar5/wg1/) includes an evaluation of
tropospheric versus stratospheric ozone using a collage of radiative transfer model calculations.
The report shows that the radiative forcing of tropospheric ozone is 10 times greater than that of
stratospheric ozone, even though only 10% of the atmospheric ozone resides in the troposphere.
The IPCC 2013 report (and references therein) also notes that ozone is a major surface pollutant,
and is important as the main source of OH, the primary cleanser of pollutants in the troposphere.
Measurements of ozone associated with deep convection are needed to characterize the extent of
ozone inter-relationships with tropospheric $H_2O$ and OH production, and in understanding cloud
microphysics/transport properties and resulting influence on global and regional tropospheric
ozone distributions.

Microphysics and photochemistry can be very complex for deep convective clouds.  Huntrieser
et al. (2016, and references therein) combined aircraft and cloud measurements with a model to
study ozone distributions and sources associated with deep convective clouds over the central
U.S.  Huntreiser et al. (2016) identified upward transport of lower tropospheric ozone and ozone
precursors into the upper troposphere within thick clouds.  They also showed that cloud tops
over-shoot the tropopause and inject high amounts of biomass burning pollutants (largely CO
and $NO_x$) and lightning-produced $NO_x$ into the low stratosphere, while at the same time ozone-
rich air from the low stratosphere is transported downward into the cloud anvil and surrounding





outflow regions as a dynamical response to overshooting. Some of the Geostationary
Operational Environmental Satellite (GOES) cloud tops were found to reach up to 17-18 km
altitude for these deep convective systems. Pronounced ozone-rich stratospheric air was
observed within cloud outflow regions.
The ozonesonde measurement record includes occurrences of very low to even "near-zero"
ozone concentrations in the tropical upper troposphere associated with the passing of deep
convective cloud systems (e.g., Kley et al., 1996; Folkins et al., 2002; Solomon et al., 2005).
The very low ozone values are largely attributed to convective lifting of low concentrations of
ozone from the marine boundary layer into the upper troposphere. In pollution-free oceanic
regions it is not uncommon for ozone in the marine boundary layer to be only a few ppbv due to
ozone net loss reactions involving hydrogen radicals OH and $HO_2$ (e.g., Solomon et al., 2005,
and references therein). Some studies suggest the possibility of in-cloud photochemical ozone
destruction mechanisms (e.g., Zhu et al., 2001; Barth et al., 2002; Liu et al., 2006). Vömel and
Diaz (2010) showed that improperly calibrated Electrochemical Concentration Cell (ECC)
ozonesondes led to a small measurement error (under-determination) and the near-zero upper
troposphere ozone concentrations reported in these studies. Vömel and Diaz (2010) found that
the near-zero ozone concentrations in the upper troposphere were instead about 10 ppbv and
attributed the calibration error to unaccounted variations associated with background cell
currents at launch. Vömel and Diaz (2010) indicate that the studies measuring "near-zero" ozone
were not wrong, but instead slightly underdetermined the low ozone concentrations.
The very low ozone measurements in the tropical upper troposphere in past studies were
obtained from a limited number of aircraft flights and ozonesondes at a few isolated sites in the
vicinity of, but not inside, deep convective cloud systems. Measuring ozone directly inside deep
convective clouds from ozonesondes and aircraft instruments remains an elusive task due to
extreme meteorological conditions associated with the clouds. Ziemke et al. (2009) developed a
residual "cloud slicing" method for measuring ozone volume mixing ratios within thick clouds
by combining Aura Ozone Monitoring Instrument (OMI) and Microwave Limb Sounder (MLS)
satellite measurements. For deep convective clouds, OMI provided the tropospheric cloud ozone
measurements after subtracting co-located MLS stratospheric column ozone. Their study found





large variability in the ozone concentrations in thick clouds. While very low ozone
concentrations (< 10 ppbv) in the clouds were identified in the remote Indian and Pacific Ocean
regions, concentrations greater than 60 ppbv were obtained over continental landmasses
including Africa. Ziemke et al. (2009) hypothesized that the ozone measured in thick clouds is
largely a manifestation of ozone concentrations (from low to high amounts) present in the low
troposphere/boundary layer that become transported upward by convection.

We build upon the cloud slicing work of Ziemke et al. (2009) to produce a long data record of
OMI/MLS cloud ozone measurements as that former study was limited to only a few months of
measurements from years 2005 and 2006. As with Ziemke et al. (2009), we derive ozone mixing
ratios inside tropical deep convective clouds by combining Aura OMI measurements of total
column ozone and cloud pressure with Aura MLS stratospheric column ozone. This paper is
organized as follows: Section 2 details the satellite measurements while Section 3 is an overview
of cloud slicing. Section 4 discusses validation and Sections 5-6 discuss basic characteristics and
scientific interpretations of the data. Finally, Section 7 provides a summary.

**2. Satellite Measurements.**

Our study combines Aura OMI and MLS ozone measurements with OMI aerosols and cloud
parameters (i.e., cloud pressures, radiative cloud fractions). OMI is a UV/VIS solar backscatter
spectrometer that makes daily measurements of Earth radiances and solar irradiances from 270 to
500 nm with spectral resolution of about 0.5 nm (Levelt et al., 2006). OMI scans perpendicular
to the orbit path with 60 side-scan positions and provides near-global coverage of the sunlit Earth
with a pixel size of 13 km × 24 km at nadir. Description and access to the OMI v8.5 data can be
obtained from the website http://disc.sci.gsfc.nasa.gov/Aura/data-holdings/OMI. In January
2009 a physical external optical blockage known as the "row anomaly" reduced the number of
the 60 good side-scanning row measurements to about 30-40. Scan positions 21-55 are the most
affected, with dependence on latitude and specific day. All of the OMI measurements that we
use were properly screened to exclude all data affected by the row anomaly artifact.



OMI cloud pressures and radiative cloud fractions are derived using UV-2 radiances (Vasilkov et
al., 2008). The cloud pressure from OMI is named optical centroid pressure (OCP). As shown
by Vasilkov et al. (2008), the OCP at UV wavelengths lies deep inside the clouds, often by
several hundred hPa and therefore is not a measure of true cloud top; they showed this by
comparing the OMI OCP measurements with both Cloudsat radar reflectivity profiles and
MODIS IR cloud pressures. The OCP effectively represents the bottom reflecting surface for the
OMI retrievals in the presence of clouds. The true ozone measurement from OMI is the column
amount from the top of the atmosphere down to the reflecting surface. In the presence of a
cloud, the OMI algorithm places an ozone "ghost column" climatology estimate below the OCP
reflecting surface to obtain total column ozone.

There are two OMI algorithms that determine the OCP. The first algorithm is based on $O_2$-$O_2$
dimer absorption (Sneep et al., 2008) and the second is based on rotational-Raman scattering
(RRS) that uses spectral structures in the ratio of backscattered radiance to solar irradiance,
known as the Ring effect (Joiner and Bhartia, 1995; Joiner et al., 2004; Joiner and Vasilkov,
2006). The two OMI cloud algorithms provide similar estimates of OCP for bright clouds
although there are small differences due to algorithmic and physical effects (Sneep et al., 2008).
We use the RRS cloud pressure for our study although our results would be nearly identical
using the $O_2$-$O_2$ cloud measurements. We refer to "cloud ozone" as the ozone column or ozone
mean volume mixing ratio lying between the tropopause and retrieved OCP from OMI under
conditions of deep convection. Deep convective clouds often have cloud tops at or near the
tropopause. Therefore much if not most of the tropospheric ozone measured between the
tropopause and OMI cloud pressure lie within the cloud itself rather than above the cloud top.

Aura MLS v4.2 profile ozone is included to measure fields of stratospheric column ozone (SCO).
MLS SCO is used in conjunction with OMI above-cloud column ozone each day to derive mean
column amounts and mean concentrations of ozone measured over deep convective clouds. The
MLS ozone profiles are vertically integrated in log-pressure from 0.0215 hPa down to the
tropopause to derive measurements of SCO as described by Ziemke et al. (2006, 2009). To
separate stratospheric from tropospheric ozone we similarly use the WMO 2K-km$^{-1}$ lapse-rate
tropopause pressure definition with NCEP re-analysis temperatures. Other tropopause pressure





definitions and other meteorological analyses besides NCEP could have also been used. We
included the WMO definition with NCEP for both historical reasons and consistency checking
relative to previous versions of our OMI/MLS tropospheric ozone products that used the same
NCEP tropopause. For the low latitudes in our study we expect that there would be only minor
differences in our results if we used instead a different tropopause. All MLS v4.2 retrieval
quality flags (quality, status, convergence, and precision) are properly adhered to for all of our
analyses. The MLS v4.2 measurements including data quality and quality flags are described in
the MLS data quality document http://mls.jpl.nasa.gov/data/v4-2_data_quality_document.pdf.
Recommended pressure levels for science applications with MLS v4.2 ozone are 0.0215 hPa to
261 hPa. With SCO representing column ozone from the top of the atmosphere down to the
tropopause, all tropospheric ozone measurements in our analyses are independent of any
stratospheric ozone barring possible unresolved stratospheric intrusions and unknown errors.

**3. Overview of Cloud Slicing.**

We use two cloud slicing methods to measure cloud ozone from Aura OMI and MLS
instruments. The first method is called "ensemble" cloud slicing. This algorithm was first
proposed by Ziemke et al. (2001) and combined co-located Nimbus-7 TOMS column ozone and
THIR IR cloud-top pressure. Here we combine OMI column ozone with OMI cloud pressure
(i.e., OCP). An advantage of ensemble cloud ozone is that it requires only a single instrument,
but weaknesses are noisiness and poor spatial resolution in the measurements. The second
method is a residual cloud slicing approach (Ziemke et al., 2009) that combines OCPs from OMI
with residual column ozone differences between OMI and MLS. An advantage of the residual
method is that it can yield measurements with high horizontal resolution. The cloud ozone
product that we generate comes from the OMI/MLS residual method. We use OMI ensemble
measurements only as a consistency check for the OMI/MLS residual ozone.

A schematic diagram for the ensemble cloud slicing method is shown in Figure 1. A region is
first chosen (top of figure, $5^o \times 5^o$ region shown) with all coincident measurements of above-
cloud column ozone plotted versus OCP effective cloud pressure (bottom of figure). The OCP as
noted in Section 2 may lie several hundred hPa below the cloud top, and the OMI algorithm





places a climatological ozone ghost column below the OCP to determine total column ozone.
For cloud slicing we use only the above-cloud ozone from OMI which is the true measurement.
In practice, we determine the above-cloud column ozone by subtracting the ghost column ozone
from total column ozone reported in the OMI level-2 orbital datasets.

In Figure 1 the OMI footprint scene depicted is 100% cloud filled so that the OCP deep inside
the cloud represents the bottom reflecting surface for the OMI retrieval.  In the more general
case, footprint scenes from OMI will not be 100% cloud filled and we account for this.  What we
actually use for cloud slicing in the Figure 1 schematic is an effective scene pressure ($P_{EFF}$) in
place of the OCP.  $P_{EFF}$ is derived from $P_{EFF} = P_{CLOUD} \cdot f + P_{SURFACE} \cdot (1-f)$, where $P_{CLOUD}$   is
the cloud OCP,  $P_{SURFACE}$  is the Earth surface scene pressure, and  $f$  is the OMI scene radiative
cloud fraction (Joiner et al., 2009).  We use OMI measurements for cloud slicing only when
radiative cloud fraction $f$ is greater than 0.80.  When $f$ is equal to 1.0 the calculated $P_{EFF}$ is
equivalent to OCP.  In our case for deep convective cumulonimbus clouds the cloud tops are
near tropopause level and so the derived mixing ratio is primarily an average measurement of
ozone inside the clouds.

Tropospheric ozone mean volume mixing ratio (VMR) is estimated by fitting a straight line to
the data pairs of above-cloud column ozone versus OCP over the selected geographical region.
This method was first described by Ziemke et al. (2001) and is summarized here.  Column ozone
($\Delta\Omega$) between two altitudes $z_1$ and $z_2$ is by definition the number of molecules per unit
horizontal area and is calculated by integrating ozone number density $n$ as $\Delta\Omega = \int_{z_1}^{z_2} n \cdot dz$.  Using
hydrostatic balance $\partial P / \partial z = -\rho g$ ($\rho$ is mass density, $g$ is acceleration of gravity) and assuming
an invariant acceleration of gravity for the troposphere this expression can be converted to:  $\Delta\Omega$
(in Dobson Units, DU; 1 DU = $2.69 \times 10^{20}$ molecules-m$^{-2}$) = $C \cdot \int_{P_1}^{P_2} X \cdot dP = C \cdot \overline{X} \cdot (P_2 - P_1)$,
where  $C = 0.00079$ DU-hPa$^{-1}$-ppbv$^{-1}$ and  $\overline{X}$ is ozone mean VMR in units ppbv.  It follows that
ozone mean VMR in the troposphere is $\overline{X}$ (ppbv)$= 1270 \cdot \Delta\Omega / \Delta P$, or in other words 1270
multiplied by the slope of the ensemble line fit.  The 2σ uncertainty for VMR in ppbv is
determined by multiplying the calculated 2σ uncertainty of the slope by 1270.  An estimate for



SCO can also be obtained by extrapolating the line fit to the mean tropopause pressure over the
region.  The above-cloud ozone at the extrapolated tropopause pressure, a direct estimate of
SCO, can be compared with MLS SCO to assess how well the ensemble method separates
stratospheric from tropospheric column ozone.

An example of ensemble scatter plots is shown in Figure 2 for October 5, 2008.  The left scatter
plot coincides with the region of southern Africa while the right scatter plot coincides with the
western Pacific.  Measured ozone mixing ratio is 72 ppbv over southern Africa and 10 ppbv over
the western Pacific.  The enhanced ozone over southern Africa suggests that ozone produced
from regional pollution including biomass burning, which is largest around September-October
each year in the SH, reaches the upper regions of the clouds.  However, the regional elevated
ozone over southern Africa may be caused by other sources including lightning $NO_x$, and
transport by the Walker circulation, and mixing of stratospheric air that is transported into the
troposphere in response to cloud tops overshooting the tropopause (e.g., Huntrieser et al., 2016,
and references therein).  The low ozone VMR in the western Pacific in Figure 2 is consistent
with low values measured in the vicinity of tropical deep convection by ozonesondes (e.g., Kley
et al., 1996; Folkins et al., 2002; Solomon et al., 2005; Vömel and Diaz, 2010).

Figure 3 illustrates the residual technique for measuring cloud ozone.  This method combines
OMI above-cloud column ozone and OMI OCP with MLS SCO.  For a deep convective cloud
the OCP lies well inside the cloud with a cloud top often at or near the tropopause, so that much
or most of measured tropospheric ozone lies inside the cloud rather than above the cloud top.
The relationship (Joiner et al., 2009) to derive residual cloud ozone VMR (units ppbv) is
$VMR = 1270 \cdot \left[ \Delta\Omega / (P_{EFF} - P_{TROPOPAUSE}) \right]$, where $\Delta\Omega$ is the difference (in DU) of OMI above-
cloud column ozone minus MLS SCO, $P_{TROPOPAUSE}$ is tropopause pressure (in hPa), and $P_{EFF}$ is
the effective scene pressure (also in hPa) as defined above.  The number 1270 is the same as for
the ensemble method to ensure units ppbv for VMR.

**4. OMI/MLS Residual Cloud Ozone Product: Validation and Consistency Checks.**



The validation of OMI/MLS residual cloud ozone measurements is not straightforward given the
paucity of in-cloud measurements from independent sources such as ozonesondes and aircraft.
However, as one approach similar to Ziemke et al. (2009), we can still obtain at least a
consistency check between the OMI/MLS residual cloud ozone and cloud ozone obtained from
the OMI-only ensemble method.

Figure 4 compares cloud ozone from the ensemble and residual techniques for July 2015 (left
panels) and October 2015 (right panels).  Both of these months coincide with the intense 2014-
2016 El Nino.  The panels in Figure 4 compare OMI/MLS residual cloud ozone (thick curves)
and OMI ensemble cloud ozone (asterisks).  The $5^o$S-$10^o$N latitude band was chosen because it
includes much of the ITCZ with thick clouds for theses months.  Both the ensemble and residual
cloud ozone in Figure 4 are low to near zero in the eastern and western Pacific close to the
dateline; it is conceivable that these oceanic regions coincide generally with pristine air and low
concentrations of both ozone and ozone precursors in the boundary layer.  In contrast, over a
broad region extending from the western Pacific to Indonesia the cloud ozone from both
measurements is enhanced.  The increased tropospheric ozone is due to a combination of
suppressed convection during El Nino and increases in biomass burning over Sumatra and
Borneo due to the induced dry conditions and wildfires (e.g., Chandra et al., 1998; Logan et al.,
2008).  The suppressed convection during El Nino coincides with reduced upward injection of
low ozone concentrations in the oceanic boundary layer compared to non-El Nino years, thus
contributing to anomalous increase in cloud ozone relative to non-ENSO years.  In the central
Atlantic the cloud ozone measurements are ~50 ppbv for both methods indicating higher ozone
concentrations injected into the clouds from below and in general a more polluted region
compared to the Pacific.  In the eastern Atlantic extending to the Indian Ocean / western Pacific
(i.e., ~$60^o$–$120^o$) the ensemble measurements are larger than for OMI/MLS.  The calculated ±2σ
uncertainties for the ensemble measurements are large everywhere including this broad region
and illustrate the noisy nature of the ensemble method.  Unlike measurements for the OMI/MLS
residual method, large errors in ozone for the ensemble method may originate largely from the
basic assumptions of the methodology such as uniformity of both SCO and tropospheric mixing
ratio throughout the chosen region.  In the next two sections we discuss the OMI/MLS cloud
ozone product for basic geophysical characteristics including some science results.





**5. Monthly Distributions.**

Figure 5 shows monthly-mean climatology maps of OMI/MLS residual cloud ozone derived from averaging similar months over the long record. Plotted in Figure 5 is mean VMR (units ppbv) representing average ozone concentration lying between the tropopause and OMI OCP as described in Section 3. In Figure 5 the mean mixing ratio is calculated for OCPs varying between 250 hPa and 550 hPa. The black regions in the figure indicate not enough deep convective clouds present and/or mostly clouds such as low-marine stratus clouds with OCP lying below the 550 hPa threshold.

The distributions in Figure 5 illustrate the large regional and temporal variability present in cloud-ozone. In the remote Pacific and Indian Ocean regions the values of cloud ozone are small at ~10 ppbv or less. High values reaching 70-80 ppbv are measured for landmass regions of India/east Asia, southern Africa and South America, and Australia. The high ozone is indicative of a more polluted lower troposphere/boundary layer.

Figure 6 shows climatology maps similar to Figure 5 but instead for "background" ozone mean VMR. The east-west tropical wave-1 pattern in tropospheric ozone (Fishman et al., 1990) is easily discerned year round in Figure 6 with high values ~60-80 ppbv in the Atlantic and low values ~20 ppbv in the eastern and western Pacific. According to Sauvage et al. (2007) using the GEOS-Chem Chemical Transport Model (CTM) the main source of tropospheric ozone in the tropical Atlantic on annual-mean basis comes from lightning $NO_x$ with smaller contributions from biomass burning, soils, and fossil fuels (by factors varying ~4-6). Their CTM also indicated that stratosphere-troposphere exchange (STE) accounts for less than about 5% of ozone in the tropical Atlantic and that most of the effects from $NO_x$ came from Africa. In the SH subtropics in Figure 6 there is a buildup of high ozone in August-November along all longitudes. Although the SH Atlantic maximum in Figure 6 occurs in every month year round, this feature also exhibits substantial inter-annual variability. Liu et al. (2017) combined GEOS-5 assimilated OMI/MLS ozone and Goddard Modeling Initiative (GMI) CTM simulations to quantify the





causes of the inter-annual variability (IAV) of tropospheric ozone over four sub-regions of the
southern hemispheric tropospheric ozone maximum. They found that the IAV of the
stratospheric ozone contribution is the most important factor driving the IAV of upper
tropospheric ozone even over two selected tropical regions: the tropical south Atlantic and
tropical S.E. Pacific. Emission influence on the tropospheric ozone variations at inter-annual and
long-term scale in general is much weaker compared to that from STE in the middle and upper
troposphere. The strong influence of emission on ozone IAV is largely confined to the
subtropical South Atlantic region in September at and below ~430 hPa.

**6. Time Series.**

With about 12 years of measurements from OMI/MLS we can analyze variability from monthly
to decadal timescales of the OMI/MLS residual cloud ozone and compare these changes with
background ozone.  In Figure 7 we show eight selected regions of interest for background ozone
(top) and cloud ozone (bottom) for October 2006.  For these eight selected regions we have
averaged cloud ozone and background ozone each month to generate long-record time series
starting October 2004.

Time series of the monthly background ozone and cloud ozone for the eight regions are plotted
in Figures 8 and 9.  In all of these eight panels the background ozone is plotted as the thick solid
curve while cloud ozone is the thin curve with asterisks.  Also plotted for the six landmass
regions in Figures 8-9 are time series of the OMI aerosol index (dotted blue curves).  In Figure 8
for northern Africa we include a line plot of the solar MgII UV index (blue squares) for
comparing decadal changes in ozone in all eight panels in Figures 8-9 with the 11-year solar
cycle.  In the eastern Pacific region in Figure 9 the Nino 3.4 index (blue squares) is also plotted
to demonstrate the dependence of cloud ozone variability from ENSO in this particular region.
All background ozone and aerosol time series in Figures 8-9 were flagged missing wherever (at
$1^o \times 1.25^o$ gridding) and whenever (monthly means) corresponding measurements for cloud ozone
were missing.

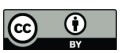



Figure 8 compares ozone time series for the following four regions: Central America, South
America, northern Africa, and southern Africa.  With the exception of the southern Africa
region, the background ozone is larger than cloud ozone by ~10-20 ppbv year round.  For
southern Africa the cloud ozone each year in summer months exceeds background ozone by ~5-
10 ppbv on average.  The annual cycle for cloud ozone with southern Africa does not appear to
be in phase with background ozone, reaching its annual maximum about 1-2 months earlier.  The
aerosol index time series in Figure 8 for southern Africa represents seasonality of biomass
burning in the region and it also peaks 1-2 months prior to maximum background ozone.
Sporadic thick clouds in the presence of tropospheric ozone from biomass burning via nearby
regions may explain the higher ozone values and 1-2 month phase lead for cloud ozone relative
to background ozone.

With Central America in Figure 8 (upper left panel) some of the month-to-month maxima and
minima for cloud ozone coincide with relative maxima and minima in background ozone on
intra-seasonal time scale.  The Central America region including the Caribbean Sea/Gulf of
Mexico and extending into the tropical north Atlantic is well documented for intra-seasonal
variability in winds and cyclonic development (e.g., Park and Schubert, 1993; Maloney and
Hartmann, 2000; Mo, 2000; Foltz and McPhaden, 2004, 2005).  Seasonal variability in Figure 8
for both background ozone and cloud ozone is most pronounced for southern Africa and weakest
for northern Africa.

For decadal time scale, the background ozone in all four regions in Figure 8 is mostly invariant
while cloud ozone shows small decreases toward the middle of the record followed by small
increases afterward.  Comparing with the MgII index in the upper right panel, this decadal
variability for cloud ozone does not appear to be directly related to the 11-year cycle in solar UV
which has minima centered around year 2009 and also at the end of the record.

Figure 9 shows time series for four additional regions: India/east Asia, Indonesia, eastern Pacific,
and Australia.  With the exception of Australia (lower right panel), the background ozone is
larger than cloud ozone by ~10-20 ppbv year round.  The cloud ozone and background ozone for
Australia are comparable during July-November months (i.e., similar to southern Africa in



Figure 8). For Indonesia and the eastern Pacific the cloud ozone is sometimes very low to even
near zero which is indicative of clean air with low concentrations of boundary-layer ozone and
ozone precursors. Indonesia in Figure 9 indicates intra-seasonal variability for both cloud ozone
and background ozone. In this western pacific region the main source of intra-seasonal
variability of tropospheric ozone is the 1-2 month Madden-Julian Oscillation (e.g., Ziemke et al.,
2015, and references therein).

Decadal changes of cloud ozone in Figure 9, with the exception of the eastern Pacific, appears
again to have relative minima around the middle of the long record and no clear connection with
the 11-year solar cycle in UV. Included in the panel for the eastern Pacific region is the Nino 3.4
index time series (squares along bottom) which was re-scaled for plotting with the ozone. For
the eastern Pacific it is clear that there is dominant inter-annual variability related to ENSO
events with associated changes in convection/SST (i.e., opposite correlation between them is
indicated). For this eastern Pacific region the cloud ozone is greatest during La Nina (suppressed
convection in the region) and lowest during El Nino (enhanced convection in the region).

It is difficult to discern timing of the seasonal minima and maxima of the aerosol and ozone time
series in Figures 8-9. For this reason we have included Figure 10 that compares 12-month
climatologies of background ozone, cloud ozone, and aerosol index time series for the six
landmass regions plotted in Figures 8-9. One main conclusion from Figure 10 is that seasonal
maxima of background ozone for the landmass regions of southern Africa, India/east Asia, and
Australia all tend to occur about one month after maxima in aerosols. For southern Africa and
India/east Asia the aerosol maximum occurs around the same month as the maximum in cloud
ozone. These phase shifts suggest that biomass burning during the mostly dry season has an
important impact on the seasonal cycles of tropospheric ozone including India where monsoon
does not generally begin until late May or early June. It is beyond the scope of our study to
explain the relative amplitude differences and phase shifts between background and cloud ozone
measurements. Explaining these characteristics will require a future investigation using either a
chemical transport model or a chemistry climate model with an appropriate convection scheme.

**7. Summary.**




We applied a residual technique to derive a data record (October 2004-recent) of tropospheric
ozone mixing ratios inside deep convective clouds in the tropics and subtropics from OMI/MLS
satellite measurements.  This residual technique makes use of the cloud optical centroid pressure
(OCP) obtained from the effects of rotational-Raman scattering in the OMI UV spectra.  Solar
UV penetrates deep into thick clouds, often by several hundred hPa.  In addition, deep
convective clouds have high cloud tops often near or at tropopause level.  As a result the
OMI/MLS cloud ozone measurements are largely indicative of ozone concentrations lying inside
the clouds.

The OMI/MLS residual cloud ozone was compared with OMI/MLS near clear-sky ozone
(denoted "background" ozone) indicating substantially lower concentrations (by ~10-20 ppbv)
for cloud ozone year round, with the exception of southern Africa and Australia during July-
November months.  For both southern Africa and Australia the seasonal maxima of cloud ozone
was found to exceed seasonal maxima of background ozone by about 5-10 ppbv.  For both
southern Africa and India/east Asia the seasonal maxima for both OMI aerosols and cloud ozone
occurs about 1-2 months earlier than for background ozone.  The analyses imply a cause and
effect relation between boundary layer pollution and elevated ozone inside thick clouds over
land-mass regions including southern Africa and India/east Asia.

While large cloud ozone concentrations ~60 ppbv or greater occur over landmass regions of
India/east Asia, South America, southern Africa, and Australia, very low cloud ozone is
persistent over the Indian Ocean and eastern/western Pacific Ocean with values ~10 ppbv or
smaller.  A low concentration of cloud ozone measured in these oceanic regions is indicative of
generally pristine air with small amounts of ozone and ozone precursors in the marine boundary
layer/low troposphere.

There is indication of intra-seasonal variability in cloud ozone over the eastern and western
Pacific Ocean regions and also over Central America.  In the western Pacific the intra-seasonal
variability originates largely from the 1-2 month Madden-Julian Oscillation.  In the eastern
Pacific the largest variability is inter-annual and originates from ENSO and associated changes




in SST/convection.  In the eastern Pacific the highest cloud ozone occurs during La Nina
(suppressed convection over the region) with lowest cloud ozone during El Nino (enhanced
convection).

Understanding changes in convection versus changes in emissions and how they relate to the
variabilities in measured cloud ozone is beyond the scope of our study.  A photochemical model
involving deep convective clouds would be necessary to study the variability for cloud ozone
from monthly to decadal time scale.  Strode et al. (2017) is current work in progress that
combines these OMI/MLS measurements with a chemistry-climate model to evaluate properties
of cloudy versus clear-sky background ozone.

The monthly gridded cloud ozone and background ozone data can be obtained via anonymous ftp
from the following:

> ftp jwocky.gsfc.nasa.gov
> Name: anonymous
> Password: (your email address)
> cd pub/ccd/data_monthly
> get vmr_30s_to_30n_oct04_to_apr16.sav



**Acknowledgments.**  The authors thank the Aura MLS and OMI instrument and algorithm teams
for the extensive satellite measurements used in this study.  OMI is a Dutch-Finnish contribution
to the Aura mission.   Funding   for   this   research   was   provided   in   part   by   NASA
NNH14ZDA001N-DSCOVR.







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












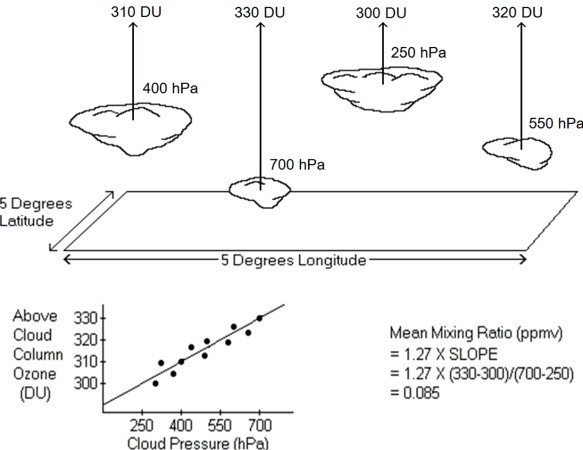


**Figure 1.** A schematic diagram illustrating the ensemble cloud slicing method involving
coincident measurements of above-cloud column ozone and cloud pressure to measure mean
volume mixing ratio (see text). For deep convective cumulonimbus clouds the cloud tops are
near the tropopause and so the mean volume mixing ratio is primarily a measurement of average
"in-cloud" ozone concentration. This figure was adapted from Ziemke et al. (2001).





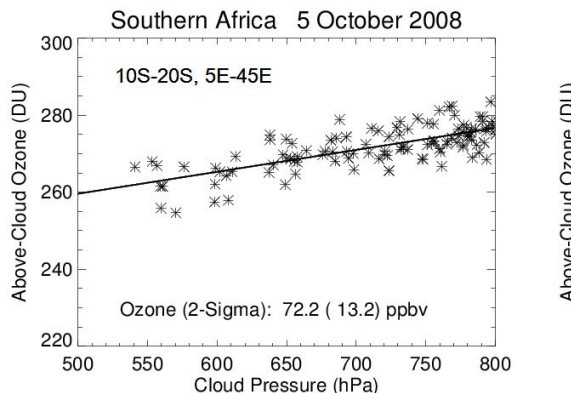
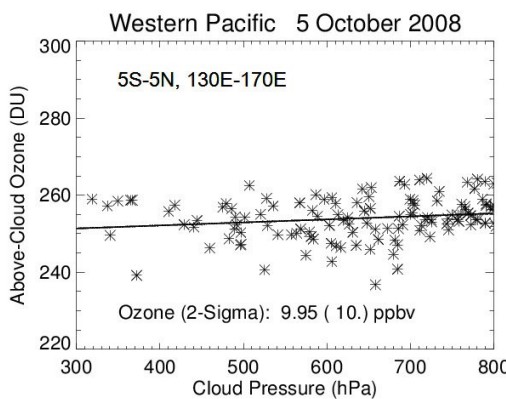


**Figure 2.** Examples of the ensemble cloud slicing technique using OMI measurements of above-cloud column ozone and cloud pressure (see text).
















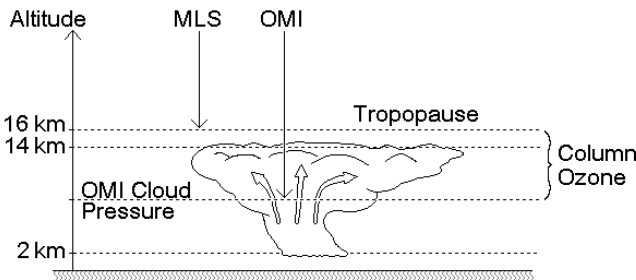


**Figure 3.** Schematic diagram of the OMI/MLS residual cloud slicing method. This depiction
shows that deep convective clouds have OMI cloud optical centroid pressures (OCPs) lying deep
inside the clouds with cloud tops often at tropopause level or very close to the tropopause. This
figure was adapted from Ziemke et al. (2009).













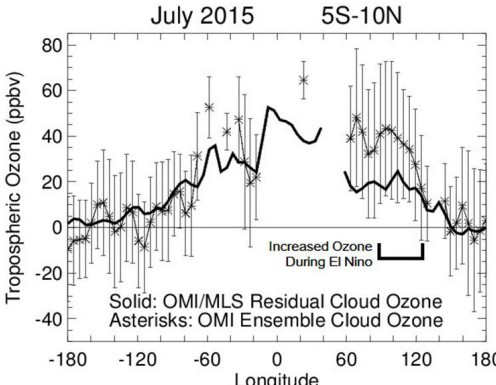
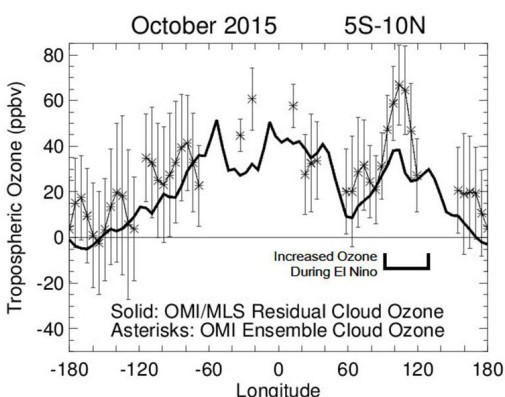

**Figure 4.** Comparisons of OMI/MLS (solid) and OMI ensemble (asterisks) cloud ozone VMR for July and October 2015 with both months coinciding with the intense 2014-2016 El Nino event. Measurements are averaged over the 5°S-10°N latitude band as a function of longitude (at 5° increments). The ensemble measurements include calculated ±2σ uncertainties. Mean VMR for the ensemble measurements are calculated for all OCP's lying between 250hPa and 550 hPa and radiative cloud fractions > 80%.

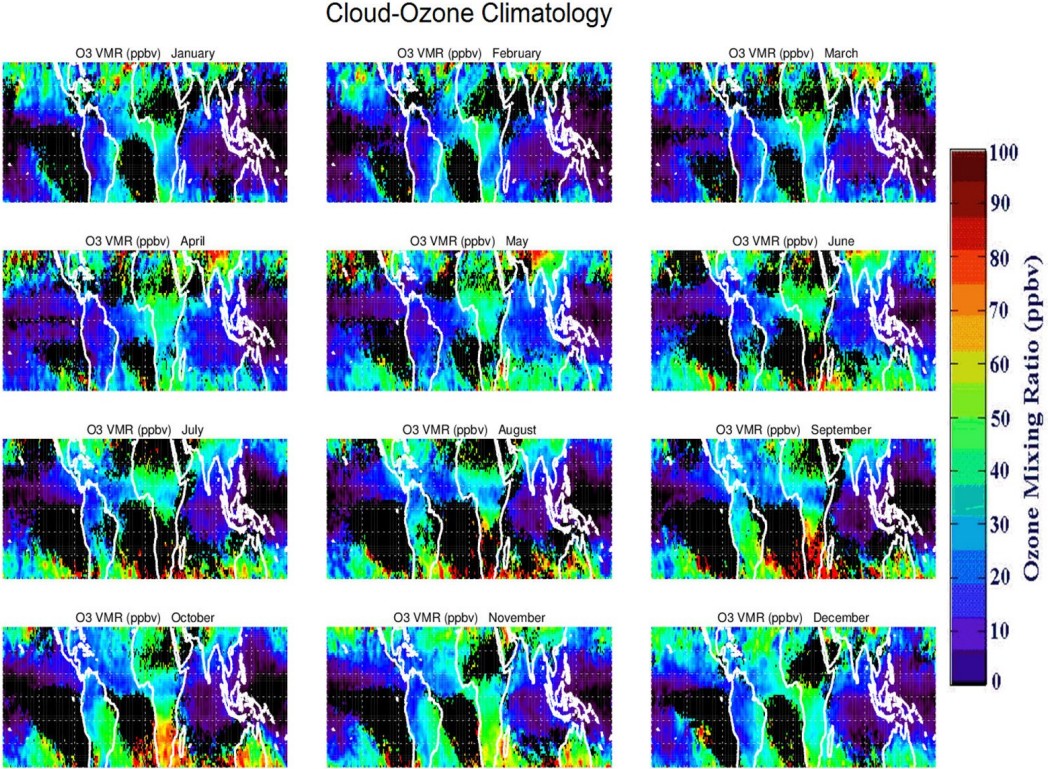

637

**Figure 5.** **M**onthly-mean climatology maps of OMI/MLS residual cloud ozone (units ppbv). Plotted is mean VMR representing average ozone concentration lying between the tropopause and OMI UV cloud pressure (OCP) as described in Section 3. The mean mixing ratio is calculated for OCPs varying between 250 hPa and 550 hPa. Black regions indicate not enough deep convective clouds present or mostly low clouds such as marine stratus clouds with OCP lying below the 550 hPa threshold.

644



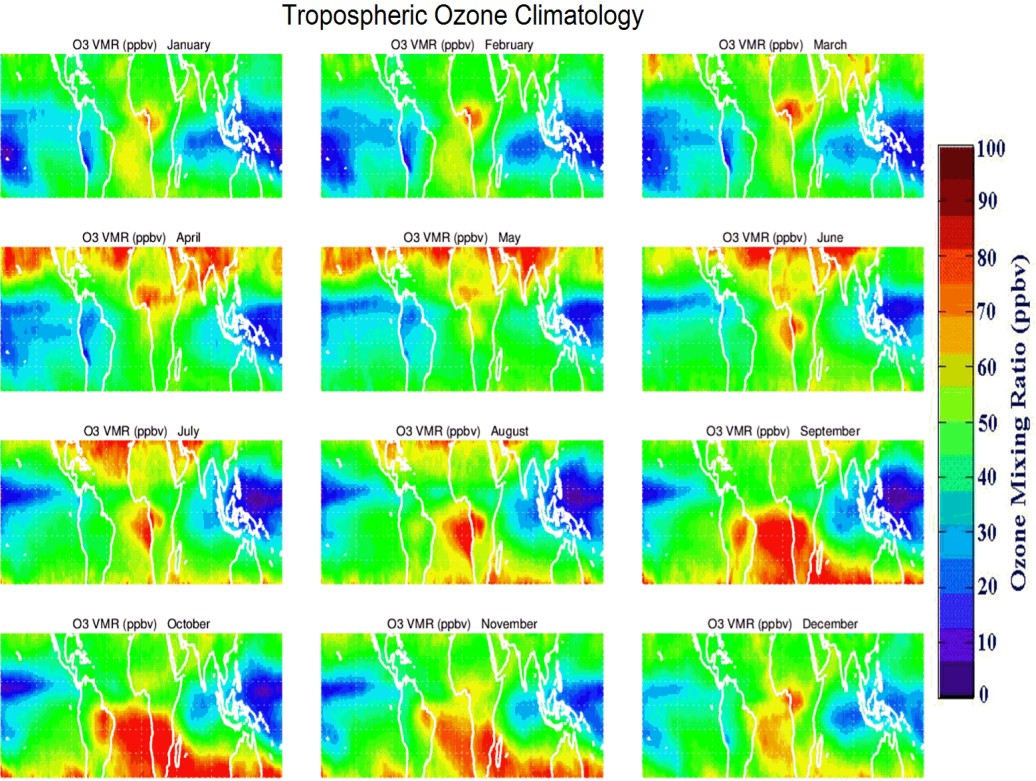

**Figure 6.** Similar to Figure 5, but instead plotting monthly-mean climatology maps of OMI/MLS VMR (units ppbv) for OMI near clear-sky scenes (i.e., radiative cloud fractions less than 30%).




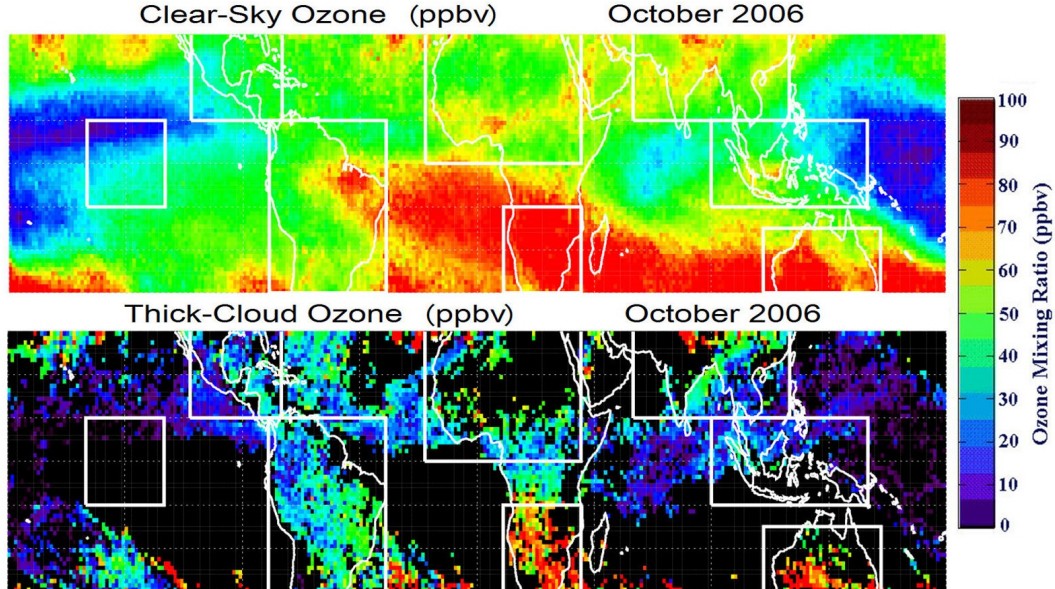

**Figure 7.** (Top) Background (near clear-sky) tropospheric ozone in units ppbv for October 2006. Shown as white rectangles are eight selected regions of interest where measurements are averaged each month to generate long record time series for October 2004 – April 2016. (Bottom) Same as top but instead for cloud ozone.




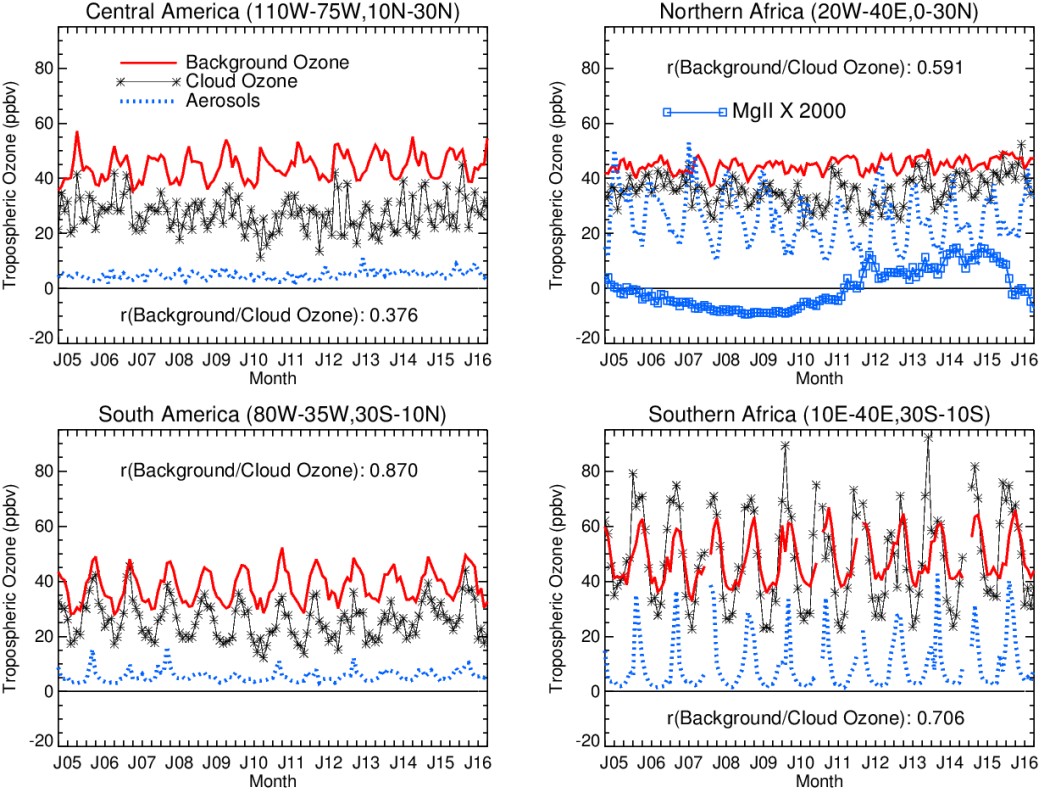


**Figure 8.** Monthly time series of background ozone (thick solid red curves) and cloud ozone (thin black curves with asterisks) for the regions of Central America, South America, northern Africa, and southern Africa in Figure 7. All ozone units are ppbv. Also shown for each of these landmass regions is the OMI monthly aerosol index time series (dotted blue curves, no units) which was re-scaled (i.e., multiplied by 60) for plotting. Included for the northern Africa region is the solar MgII index (SI units) that has been re-scaled for plotting (i.e., average removed and then multiplied by 2000).

665



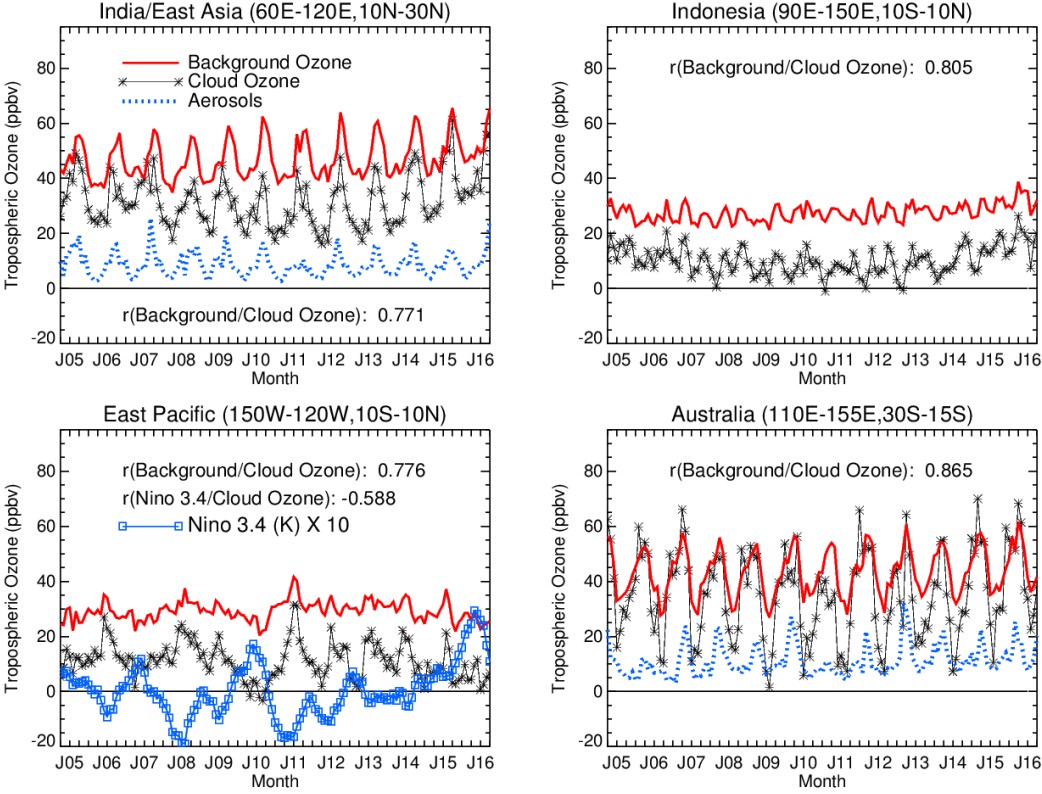

666

**Figure 9.** Similar to Figure 8, but instead for the regions of India/east Asia, Indonesia, eastern Pacific, and Australia. Aerosol index time series (dotted) for the landmass regions is again shown. Also included for the eastern Pacific (lower left panel) is the Nino 3.4 index (blue squares, units K) and its correlation with cloud ozone. The Nino 3.4 index was re-scaled (multiplied by 10) for plotting with ozone time series.

672





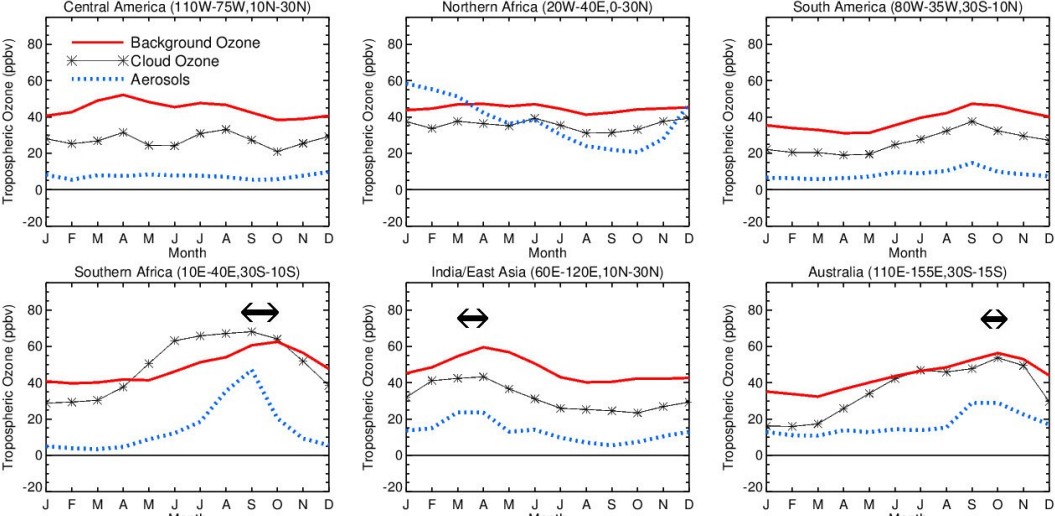

673

**Figure 10**. Twelve-month climatology time series for the six continental land-mass regions
plotted in Figures 8 and 9 using the same color scheme. Shown here are background ozone
(solid red curves), cloud ozone (asterisks), and aerosol index (dotted blue curves). The OMI
aerosol index has been re-scaled (i.e., multiplied by 60) for plotting. Approximate phase shifts
between background ozone and aerosol index time series are shown with dark arrows.

679