# Peer review of "A Cloud-Ozone Data Product from Aura 1 OMI and MLS Satellite Measurements 2 3 4 Jerald R. Ziemke1,2, Sarah A. Strode2,3, Anne R. Douglass2, Joanna Joiner2, Alexander 5 Vasilkov2,4, Luke D. Oman2, Junhua Liu2,3, Susan E."

_Atmospheric Measurement Techniques, 2017_

## Referee Comment (RC1) · Anonymous Referee #2 · 29 May 2017

General Comments:

This paper describes a long-term record of monthly mean cloud ozone in the tropics derived from OMI/MLS measurements. It is derived from the residual between OMI ozone above convective clouds and MLS stratospheric ozone. As the OMI cloud pressure is often in the middle of clouds and the physical cloud tops often reach the tropopause, the derived residual ozone generally therefore represents ozone inside clouds, difficult to be measured otherwise. The OMI/MLS cloud ozone is also compared with cloud ozone derived from OMI alone using the ensemble cloud slicing. The spatiotemporal distribution of cloud ozone is discussed in contrast with background ozone. The analysis shows persistent low ozone in the tropical Pacific and higher ozone over landmass regions and connections with ENSO, intra-seasonal/Madden-Julian oscillation variability and boundary layer pollution. This study is suitable for publication in AMT. It is

well logically organized. Overall, I recommend it to be published after addressing the following minor comments.

Specific Comments:

1. L115, do you mean OMI V3 as V8.5 is for the OMTO 3algorithm not for all OMI products

2. L251, the sentence "The panels in Figure 4 . . . ozone (asterisks)" is redundant with the first sentence and can be removed.

3. L267-269, you may add something to explain the larger uncertainty, e.g., due to the sparseness of clouds as indicated by much fewer derived ensemble cloud ozone in this region

4. Last paragraph of section 3, is the OMI/MLS cloud ozone product derived on the daily basis? If not, mention monthly mean and the grid cell for averaging. Briefly mention that it is limited to 30S-30N and explain why.

5. L286-290, please mention the enhanced ozone over the Pacific/Atlantic Ocean at latitude closer to 30S/30N.

6. L292, it is good to define "background" here, i.e., by adding "(near clear-sky scenes with radiative cloud fractions less than 30%)"

7. In last paragraph of section 5, is it contradictory between saying "STE accounts for <5% of ozone over tropical Atlantic" around L299 and "stratospheric ozone contribution is the most important factor for driving the IAV of upper tropospheric ozone . . ." around L308? Please clarify it.

8. In Figure 8 and L335-345, it is useful to add correlation between aerosol index and cloud ozone over Southern Africa

9. L356-360, any other speculation for the relatively low cloud ozone around 2010-2012?

Technical comments

1. L19, change to "tropospheric" as we say "lower tropospheric ozone"

2. L25, use full word "ultraviolet" or "ultraviolet (UV)" for UV at its first occurrence

3. L37 and other places, remove the "." at the end of section title.

4. L111, change to "UV/visible" due to first/only occurrence of visible

5. L144, change to "lies"

6. L171, please define for "THIR" and "IR" at its first occurrence

7. L182, remove "effective cloud pressure" as OCP stands for cloud pressure

8. L253, please define ITCZ at its first occurrence.

---

## Author Comment (AC1) · 9 Aug 2017

General comments:

This is a generally well written manuscript on the retrieval of tropospheric ozone within convective clouds retrieved from UV nadir observations with the OMI instrument complemented by MLS observations of stratospheric ozone. I think the manuscript should eventually be published, but there are several aspects that should first be addressed in my opinion.

The approach used to determine what is called "cloud ozone" is quite pragmatic. This is not necessarily a problem, but the limitations of the applied method are not discussed in sufficient detail in my opinion.

A problem with validating the OMI/MLS cloud ozone is lack of independent ozone measurements for deep convective clouds. The Ziemke et al. (2009) paper was a predecessor to the current paper and included validation and discussions of limitations of the cloud ozone measurements and also included RT model calculations of vertical sensitivity of ozone in deep convective clouds. The Strode et al. (2017) paper (currently in review and listed in the references) is a related side study. Strode et al. (2017) tests the validity of the OMI/MLS residual cloud ozone measurements against a freerunning chemistry climate model (CCM). The CCM is found to simulate key features of both the cloudy-clear differences and the geographic distribution of the in-cloud ozone from OMI/MLS. In our study we include an anonymous ftp site for the cloud ozone data. We look forward to getting feedback from other researchers on the usefulness and quality of these measurements from their analyses.

It is stated several times that the derived cloud ozone corresponds to the average O3 VMR inside the cloud. However, the nadir measurements are probably very insensitive to O3 in the lower or middle part of a convective cloud, i.e. the retrieved O3 VMR reflects O3 in the upper part of the cloud and does that in a non-trivial way, probably. In this respect it would be very valuable to determine and/or show a measure of the sensitivity of the retrieval to O3 at different levels below cloud top. Perhaps you have already done sensitivity studies like that for earlier papers?

I'm also wondering how different the cloud penetration depths at the wavelengths used for the OCP and the O3 retrievals are. The wavelengths are quite close, so the difference is probably not too large, but it may affect the results in a non-trivial way.

I'm also wondering, what the effect of light-path enhancements due to multiple scattering inside the clouds on the O3 retrievals is? The RT is quite complex in this case and I'm not sure, whether this complexity can simply be neglected.

It is mentioned several times that the OCP is deep within the cloud (several 100 hPa below the actual cloud top). This surprises me and I wonder, whether this is expected.

Have you performed simulations of the RT inside the cloud? The fact that OCPs are well below the cloud top suggests that a large fraction of the UV photons can penetrate the cloud deeply. I'm not sure this is expected. Perhaps I'm missing a point here. Please add more information here and, if available, mention or cite studies that deal with this complex RT problem.

You have very valid comments above regarding the issue of UV penetration in thick clouds. We have actually done a considerable amount of work on addressing the issues that you mention above with several papers listed in the references. The Vasilkov et al. (2008), Ziemke et al. (2009), and Joiner et al. (2012) papers may be the most detailed and included a comprehensive RT code that includes the effects of multiple scattering within clouds (denoted LIDORT-RRS). We discuss these papers including the Vasilkov et al. (2008) paper further below pertaining to your comments.

You often use the term "above-cloud column", which is misleading, because the column in the paper actually also includes the ozone in the top part of the clouds. I suggest using another term or at least emphasizing this point explicitly in the paper.

In section 2 (third paragraph) we had mentioned in a sentence that we refer to the "cloud ozone" as the ozone column or mean VMR lying between the OMI OCP and the tropopause. In our revision right after that sentence we now add that we refer to "above-cloud ozone" as the column ozone measured from the top of the atmosphere down to the OMI OCP. We double checked to make sure that we didn't accidently type "above-cloud ozone" for "cloud ozone" (or vice versa) in the paper.

I would like to point out that my intention is not to ask you to do a lot of RT simulations (perhaps you have already done so, though) to address the issues raised above, but rather to discuss these aspects openly (you've probably thought about all of them, and perhaps they are not that important), and to discuss the limitations of the method and the results.

Specific comments:

Line 56: "Huntreiser" -> "Huntrieser"

Done.

Line 123: "As shown by Vasilkov et al. (2008), the OCP at UV wavelengths lies deep inside the clouds, often by several hundred hPa and therefore is not a measure of true cloud top;" I'm surprised that the OCP is so much below the cloud top at UV. Is this expected based on the approach to estimate cloud top pressure using the OMI UV radiances?

Thanks for the comment – this is an important property of the UV measurements. In section 2 and in the references we included several papers by Joiner and Bhartia (1995), Joiner et al. (2004), Joiner and Vasilkov (2006), Vasilkov et al. (2008), Ziemke et al. (2009), and Joiner et al. (2012) on this subject. In the revision we added the Joiner et al. (2012) paper as it is more recent than the others. The papers by Vasilkov et al. (2008), Ziemke et al. (2008), Ziemke et al. (2009), and Joiner et al. (2012) used a Linearized Discrete-Ordinate Radiative Transfer RRS (LIDORT-RRS) code (Spurr et al., 2007) for calculations. They studied sensitivity to geometrical cloud thicknesses and included CloudSat reflectivity profiles and MODIS IR cloud pressures. They describe the large differences (up to several hundred hPa) between physical cloud top and OCP measured by OMI using their OMCLDRR algorithm. Below is Figure 12 adapted from the Vasilkov et al. (2008) paper that illustrates this result:

Figure 12. Cloudsat radar reflectivity on 13 November 2006 with cloud pressures retrieved from OMCLDRR (rust triangle curves, denoted UV), retrieved from MODIS (top red squares, denoted IR), and simulated for OMI on the basis of CloudSat/MODIS data (black diamonds).

The Ziemke et al. (2009) and Joiner et al. (2012) papers listed in the references and main text also involved further LIDORT-RRS calculations. The Joiner et al. (2012) paper describes a fast simulator to compute the OCP quickly from profiles of optical extinction such as from models or CloudSat/MODIS.

Line 137: "for bright clouds". What about clouds that are not "bright"? I'm wondering how one would distinguish between bright clouds and the other ones. Do you only use bright clouds in this study?

We only used bright cloud scenes to derive all of the cloud ozone measurements – the bright cloud scenes for the residual and ensemble methods both used only OMI scenes with radiative cloud fraction greater than 80%.

Line 140: I find the term "cloud ozone" somewhat misleading, because it certainly does not correspond to the entire ozone column inside the cloud. The OCP will generally be well above the cloud bottom and the "cloud ozone" will then correspond only to a fraction of the column ozone actually inside the cloud.

You are correct that we don't measure ozone mean VMR for the entire thick cloud but instead between the tropopause and OMI OCP. In section 2 we mentioned that the "cloud ozone" from OMI/MLS refers to the ozone mean VMR lying between the OCP and tropopause. We have now also added in the revision that "above-cloud ozone" refers to the column ozone from the top of the atmosphere down to the OCP measured by OMI.

Line 147: "OMI above-cloud column ozone". This also includes the "cloud ozone", right? I think this should be mentioned explicitly, because for the inexperienced reader this is not obvious, and it may suggest that there are different OMI ozone column data products.

This is related to the previous comment where we added the definition for above-cloud ozone which should help clarify the discussion in this paragraph.

Line 162: "With SCO representing column ozone from the top of the atmosphere down to the tropopause, all tropospheric ozone measurements in our analyses are independent of any stratospheric ozone barring possible unresolved stratospheric intrusions and unknown errors." I don't agree with this statement. An important aspect is the

(limited) vertical resolution of the MLS ozone profiles. MLS will not be able to retrieve the true vertical variation of ozone, but the measurement process corresponds to (roughly speaking) the convolution of the true vertical ozone profile with the MLS O3 averaging kernels, which will have a width of several km. This means, that some of the stratospheric O3 may (or rather will) be smeared into the troposphere. This effect will probably be on the order of at least several DU, potentially significantly more. Perhaps this aspect has been addressed in previous studies already?

Very good point... The MLS v4.2 data quality document shows that the vertical resolution for MLS about the tropopause is  $\sim$ 3 km which will affect the quality of the cloud ozone measurements from OMI/MLS. In the revision we have deleted those sentences and replaced with new ones mentioning inherent errors in SCO from both NCEP tropopause pressure and MLS retrieval errors (esp. vertical sensitivity about the tropopause).

Figure 1, line 592: "For deep convective cumulonimbus clouds the cloud tops are near the tropopause and so the mean volume mixing ratio is primarily a measurement of average "in-cloud" ozone concentration." I don't think this statement is correct. I agree that for well developed clouds one can assume that their tops are close to the tropopause, but the fraction of the measured column below cloud top will certainly not correspond to the average ozone amount inside the cloud, right? You measurement will be rather insensitive to the amount of ozone in the lower part of the cloud. It would be interesting to know what the mean penetration depth of UV radiation at around 350 nm inside optically thick clouds is.

This relates to the comments above for Line 140 including RT calculations by Vasilkov et al., (2008).

Figure 1: y-axis label of the inset: "above cloud column ozone". I think this is misleading (or I'm missing the point), because this ozone columns includes your "cloud ozone", right?

Thanks – we have revised the Figure 1 caption and discussion to clarify this point.

Line 182: "above cloud column ozone". See last comment.

Relating to previous comment.

Line 192: Effective scene pressure. I'm wondering, whether it would be better to determine an effective scene altitude, rather than pressure. But if you use only cases with f > 0.8 this probably does not make a big difference. Perhaps there was a specific reason to use pressure here?

We chose to use pressure since the OMI ozone algorithm and measurements are based on vertical pressure coordinate. Other instruments like OMPS or SAGE as examples measure number density as a function of altitude – for those measurements it may be more advantageous to use altitude rather than pressure.

Line 197: "In our case for deep convective cumulonimbus clouds the cloud tops are near tropopause level and so the derived mixing ratio is primarily an average measurement of ozone inside the clouds." As mentioned above, I don't think this is true. I think the derived mixing ratio is some sort of average over a part of the cloud, and it's probably non-trivial to determine what part of the cloud this actually is. Again, if you know what the estimated penetration depth is, this would be a useful piece of information. Also, as mentioned above, the light-path enhancement due to multiple scattering will affect the sensitivity of the measurements to ozone inside the cloud.

It is difficult to quantify the penetration depth relative to actual physical cloud top. To get cloud-top information we need to use other ancillary cloud measurements that have different view times and are not properly co-located. Making 1-1 evaluations even more difficult is that these clouds including their cloud properties can change relatively quickly on time scales of just ~10-15 minutes. Having UV and IR sensors on the same spacecraft and with near-equivalent sample times would be nearly ideal for getting at coincident OCP and IR cloud tops and estimated penetration depth. Actually, what

is really needed is a combination of an imager like MODIS and CloudSat to provide optical depth profiles. For OMI measurements we filter thick clouds for OCPs < 550 hPa and anticipate (as suggested in the figure above from Vasilkov et al.) that these are generally optically thick clouds with physical tops near the tropopause. In previous papers we addressed multiple scattering including multiple cloud decks (also illustrated in this same figure at the far left) – resulting errors are largely reduced from the RCF and OCP filtering that we apply to the scene data.

Line 253: "theses" -> "these"

Done.

Line 628: "OCP's" ->"OCPs"

Done.

Figures 8 and 9: I'm not sure, how robust the differences between background and cloud ozone really are. I accept that the clear sky values are probably very realistic average tropospheric O3 VMRs, but I'm not sure the cloud ozone is really a good measurement for the O3 VMR inside the cloud. There must be differences – perhaps small – in the penetration depths at the wavelengths used for the O3 retrieval and the OCP retrieval. This may lead to systematic errors. And I'm not sure, whether the light-path enhancements inside the cloud are compensated entirely by using the OCP retrievals for reference. These aspects should be commented upon, I think. The paper is still interesting, but I think the limitations of the technique should be stated. And if all of these potential problems are well understood – i.e. no limitations – this should also be mentioned.

Differences between the clear and cloudy ozone time series in Figures 8-9 are definitely robust in terms of sheer magnitudes and seasonal variability. These differences seem to make sense under hypothesis of injection of (high to low) concentrations of boundary layer ozone upward into the clouds. You also mention about the differences of penetration depths between the retrievals for ozone and OCP. It turns out that the difference is about 317-330 nm for ozone and 350 nm for OCP. Most of the scattering inside the clouds is from cloud particles and not from Rayleigh. Our simulations show only small differences in photon path versus wavelength. In the referenced papers we evaluated such potential errors/limitations as you have mentioned above. As note, there are going to be systematic errors, both regional and temporal in nature with the cloud ozone measurements due to so many parameters involved. These include errors in retrieved OMI OCP and column ozone, errors with MLS ozone profiles, and errors for even just one of these quantities is extremely difficult and likely not possible to accomplish with useful degree of accuracy.

Please also note the supplement to this comment: https://www.atmos-meas-tech-discuss.net/amt-2017-107/amt-2017-107-AC1supplement.pdf

---

## Author Comment (AC2) · 9 Aug 2017

General Comments:

This paper describes a long-term record of monthly mean cloud ozone in the tropics derived from OMI/MLS measurements. It is derived from the residual between OMI ozone above convective clouds and MLS stratospheric ozone. As the OMI cloud pressure is often in the middle of clouds and the physical cloud tops often reach the tropopause, the derived residual ozone generally therefore represents ozone inside clouds, difficult to be measured otherwise. The OMI/MLS cloud ozone is also compared with cloud ozone derived from OMI alone using the ensemble cloud slicing. The spatiotemporal distribution of cloud ozone is discussed in contrast with background ozone. The analysis shows persistent low ozone in the tropical Pacific and higher ozone over landmass regions and connections with ENSO, intra-seasonal/Madden-Julian oscillation variability and boundary layer pollution. This study is suitable for publication in AMT. It is well logically organized. Overall, I recommend it to be published after addressing the following minor comments.

Specific Comments:

1. L115, do you mean OMI V3 as V8.5 is for the OMTO3 algorithm not for all OMI products

We now inserted a sentence to clarify that v8.5 is the actual retrieval algorithm for the OMI ozone.

2. L251, the sentence "The panels in Figure 4 : : : ozone (asterisks)" is redundant with the first sentence and can be removed.

We had a typo – the parentheses in the first sentence should have been singular stating as "panel" rather than "panels". The third sentence states that OMI/MLS residual cloud ozone is represented by the thick curve and ensemble cloud ozone is represented by the asterisk curve in each of the two panels.

3. L267-269, you may add something to explain the larger uncertainty, e.g., due to the sparseness of clouds as indicated by much fewer derived ensemble cloud ozone in this region

Done.

4. Last paragraph of section 3, is the OMI/MLS cloud ozone product derived on the daily basis? If not, mention monthly mean and the grid cell for averaging. Briefly mention that it is limited to 30S-30N and explain why.

These are excellent points. . . We have now clarified the use of daily measurements that were then averaged monthly in the first paragraph, and also the final two paragraphs

that descry be the OMI ensemble method and OMI/MLS residual method. We mention at the end of section 3 why we limited to 30S-30N regarding noise issues.

5. L286-290, please mention the enhanced ozone over the Pacific/Atlantic Ocean at latitude closer to 30S/30N.

Good point. We have now included discussion of the observed cloud ozone concentrations over ocean in this paragraph.

6. L292, it is good to define "background" here, i.e., by adding "(near clear-sky scenes with radiative cloud fractions less than 30%)"

Another good point. . . Done.

7. In last paragraph of section 5, is it contradictory between saying "STE accounts for <5% of ozone over tropical Atlantic" around L299 and "stratospheric ozone contribution is the most important factor for driving the IAV of upper tropospheric ozone : : :" around L308? Please clarify it.

Thanks for catching this – although both Sauvage et al. (2007) and Liu et al. (2017) examined the tropospheric ozone over the tropical Atlantic, Sauvage et al. (2007) focused on the source contribution of tropospheric annual-averaged ozone budget. In contrast the Liu et al. (2017) conclusion was focused on the source contribution of ozone IAV during the austral winter season in the middle and upper troposphere, of which there are large ozone changes due to STE. We have rewritten the text to make this clear.

8. In Figure 8 and L335-345, it is useful to add correlation between aerosol index and cloud ozone over Southern Africa

Done.

9. L356-360, any other speculation for the relatively low cloud ozone around 2010-2012?

It could instead be related partly to ENSO decadal variability (e.g. Nino 3.4 plotted in following Figure 9 for the east Pacific) but that is also speculation. We would really need a longer record and a comprehensive stratosphere-troposphere photochemical transport model to attempt to explain.

Please also note the supplement to this comment:
https://www.atmos-meas-tech-discuss.net/amt-2017-107/amt-2017-107-AC2-supplement.pdf

---

## Author Response (AR2)

The authors generally did a good job responding to my previous comments. I only have the following minor comments (the page numbers refer to the version with highlighted changes):

- Page 14, last paragraph before section 3: The vertical resolution issue is very likely a systematic issue, and will not cancel out if a large number of measurements are averaged. The reason is that there will essentially always be a large increase in O3 above the tropopause, i.e. O3 will be "smeared into" the troposphere by the MLS inversion procedure and lead to an overestimation of O3 below the thermal tropopause.

This is a good point that you mention regarding the vertical resolution of MLS. We have added extra text in the revision in section 2 to discuss this issue further. The smearing of retrieved ozone between troposphere and stratosphere is really a main reason that we use MLS.

Below is a key figure from the MLS v4.2 ozone data quality document. The integrated AKs (thick solid curve) and FWHM estimated vertical resolution (thick dashed curve) attest to the exceptional ability of MLS to derive stratospheric ozone columns. About the tropopausee MLS has an estimated vertical resolution of ~3 km in this figure. This resolution is very good when compared to other current instruments for isolating stratospheric columns, particularly the nadir profilers like SBUV (or TES, IASI, etc.) that have vertical resolution about the tropopause ~10 km or larger. There may indeed be a systematic bias in the OMI/MLS cloud ozone due to the MLS uncertainty about the tropopause, but we can't quantify this number even if it does indeed exist (although we would certainly correct for it if we could first identify it). We mention all of this now in the revision including the possibility of several DU systematic error in the cloud-ozone measurements.

[Figure]

(This figure is from the MLS v4.2 ozone data quality document)

- Page 18, section 5, line 5: "between 250 hPa and 550 hPa"
Why is only such a limited range of OCP used for the analysis? Are there issues if OCP values outside this range are used? Please comment on this in the paper.

Thanks for pointing this out… In the revision in section 5 we now discuss our choice of using 250 hPa < OCP < 550 hPa.   In principle this range for OCP helps ensure high deep convective clouds with physical cloud tops generally at or near tropopause level (relates to the OCP results including Figure 12 of Vasilkov et al., 2008).

- Caption Fig. 8, last line: "include" -> "included"

Thanks - done.